# Graph-based Confidence Calibration for Large Language Models

## Abstract

One important approach to improving the reliability of large language models (LLMs) is to provide accurate confidence estimations regarding the correctness of their answers. However, developing a well-calibrated confidence estimation model is challenging, as mistakes made by LLMs can be difficult to detect. We propose a novel method combining the LLM's self-consistency with labeled data and training an auxiliary model to estimate the correctness of its responses to questions. This auxiliary model predicts the correctness of responses based solely on their consistent information. To set up the learning problem, we use a weighted graph to represent the consistency among the LLM's multiple responses to a question. Correctness labels are assigned to these responses based on their similarity to the correct answer. We then train a graph neural network to estimate the probability of correct responses. Experiments demonstrate that the proposed approach substantially outperforms several of the most recent methods in confidence calibration across multiple widely adopted benchmark datasets. Furthermore, the proposed approach significantly improves the generalization capability of confidence calibration on out-of-domain (OOD) data.

## 1 Introduction

In recent years, large language models (LLMs) have demonstrated remarkable capabilities across various natural language processing tasks such as question answering (Wei et al., 2022; Shen et al., 2023; Zheng et al., 2023; Qin et al., 2023; Singhal et al., 2023), text summarization (Tang et al., 2023; Deroy et al., 2023; Tam et al., 2023; Roit et al., 2023), and even creative writing (Gómez-Rodríguez & Williams, 2023; Wang et al., 2024; Deng et al., 2024). Despite their impressive performance, LLMs often give wrong answers in question-answering tasks. There is an urgent need to check the correctness of LLMs' responses. One particularly interesting question is to calibrate the confidence levels of the correctness of responses from LLMs (Kuhn et al., 2022; Ulmer et al., 2022; Van Landeghem et al., 2022; Vazhentsev et al., 2023; Ulmer et al., 2024). Accurate confidence estimation is vital for deploying LLMs in the real world, as it allows users to gauge the reliability of the model's predictions and make informed decisions based on these outputs. On the contrary, miscalibrated confidence may lead to over-trust in incorrect responses or doubts about the correct ones. For example, a misleading response may steer a patient in the wrong direction when making health decisions; it may also lead an investor to impulsive investment decisions.

In this work, we consider calibrating the confidence with the correctness of LLMs' responses. This task is challenging in several aspects. First, due to LLMs' superior ability to generate text, mistakes in an LLM's response are usually at the semantic level, making it hard to detect even for humans. There are methods using an auxiliary Language Model (e.g., DeBERTa (He et al., 2020)) to check whether the LLM's response answers the question Ulmer et al. (2024). Since the LLM is supposed to be much stronger than the LM, the LLM should be able to avoid most mistakes that can be detected by an LM; this type of method may omit a significant fraction of wrong answers. Second, it is hard to detect mistakes from the LLM's internal working mechanism. Because the LLM uses many hidden layers to process the information, it is hard to discern the signal from a small number of hidden units. Even if this is possible, it is not easy to apply this type of method to black-box LLMs.

Recently, there has been some progress in quantifying the model's own confidence in a response through consistency among the model's responses (Chen & Mueller, 2023; Lin et al., 2024). If the

LLM always gives similar responses, then there is less uncertainty, and any one of these responses tends to have a high probability of being correct. In particular, the results show that the model's own confidence in its response has a strong correlation with the correctness of the response. This new development leads to an important research question: whether we can calibrate the confidence of correctness from consistency among the LLM's responses.

In this work, we develop a new method to calibrate the confidence of correctness from the consistency of an LLM's responses. To achieve this goal, we train a separate calibration model to predict the correctness of the LLM's responses. To get the input to the calibration model, we form a similarity graph over the LLM's multiple responses to the same question. The similarity graph encodes information about the consistency between LLM's responses. A response consistent with more answers tends to have a higher likelihood of being correct, so the consistency graph is predictive of the responses' correctness. In our new approach, the calibration model only considers the consistency among responses without processing any actual language information. Thus, we can use a relatively simple and efficient model.

Our model achieves premium performance in the empirical study. Compared with previous calibration methods, our model has much better calibration performance because of the usage of consistency graphs. Compared to prior methods based on consistency inputs, our method improves not only the calibration performance but also the ranking performance. Furthermore, our method improves the generalizability of confidence calibration for out-of-domain settings, demonstrating the advantage of using a separate learning model. In summary, our **main contributions** are:

- **Graph-based confidence calibration method:** We propose a novel graph-based confidence calibration approach to improve the reliability of LLMs.

- **Enhanced calibration performance:** Our evaluations demonstrate that the proposed method substantially outperforms recent methods in confidence calibration across several widely used benchmark datasets.

- **Improved OOD generalizability:** Evaluations on OOD confidence calibration show that our graph-based approach significantly improves generalizability in OOD settings.

## 2 RELATED WORK

Due to the urgent need to improve the reliability of LLMs, confidence estimation and calibration for these models have become active areas of research. Existing research in LLM uncertainty quantification can be summarized into two main categories: uncertainty quantification and confidence calibration (Geng et al., 2023). Confidence estimation for short responses (e.g., for multi-choice or yes-no questions) is generally less complicated than for long responses (Ye et al., 2024). For a brief response, the LLM's output logits are informative about its confidence; the easy comparisons of responses to the true answer facilitate both calibration and evaluation. Confidence estimation for long responses cannot simply depend on LLM's output logits (Duan et al., 2023; Bakman et al., 2024) because the logits indicate more about the probability of text and less about the semantics behind it. There are also methods using the internal state of an LLM (Ren et al., 2022), but it is not always available to have such information about the LLM interface.

Another approach is to check the LLM's consistency in its responses. Kotelanski et al. (2023) demonstrate that repeated sampling and consistency checks across multiple outputs can serve as reliable proxies for model confidence. Manakul et al. (2023) generates multiple responses from the LLM and checks the consistency between responses using various methods, including querying the LLM. Chen & Mueller (2023) combines the consistency between responses and the LLM's self-reflection certainty to quantify the uncertainty. Kuhn et al. (2022) considers confidence from semantic equivalence and proposes a method based on clustering of responses. Lin et al. (2024) organize responses in a graph with their pairwise semantic similarity and then extract graph statistics for confidence estimation. Zhang et al. (2024) examines methods of comparing responses via entailment and contradiction relationships. These studies highlight the importance of semantic consistency in confidence estimation. These methods are evaluated by comparing their estimated confidence values against the correctness of responses. The correct labels are often obtained by *reference matching* (Huang et al., 2024), checking whether responses match true answers with a particular similarity measure.

To better calibrate the confidence estimation, some methods directly use correctness labels in their calibration procedures. Mielke et al. (2022) trains a calibrator to predict the correctness of a response for a given question. With a similar idea, Ulmer et al. (2024) trains a language model (e.g., DeBERTa) based on question-response pairs to predict the probability of responses' correctness. Based on SelfCheckGPT (Manakul et al., 2023) and JAFC (Tian et al., 2023), Chen et al. (2024) train supervised models to reduce grouping losses and improve the confidence estimation. The method by Liu et al. (2024) uses an LLM's latent representations to predict the correctness of responses. Detommaso et al. (2024) uses the "multicalibration" technique to calibrate the probability of correctness. (Fadeeva et al., 2023) offers a detailed comparative study of various confidence estimation methods, providing empirical evidence on their effectiveness across different tasks. However, these studies have not sufficiently exploited response consistency to predict the probabilities of the responses being correct.

## 3 METHOD

Our ultimate goal is to quantify the probability of the correctness of a response from an LLM. Since the LLM can give a correct answer with different phrases, we need to consider the probability that the response is semantically correct.

**Background:** The formulation of semantic equivalence (Kuhn et al., 2022) provides a framework for our analysis. Let $\mathcal{R}$ be the space of all possible responses. Given a question $q$, the space $\mathcal{R}$ is divided into a set $\mathcal{C}_q$ of semantic classes: $\mathcal{R} = \cup_{C \in \mathcal{C}_q} C$ and $C' \cap C = \emptyset$ for any two different semantic classes $C, C' \in \mathcal{C}_q$. For two responses $r_1, r_2 \in C$ in the same equivalent class, they are considered as the same semantic response: if one is the correct answer, the other is correct as well, and vice versa. Then, we can consider the quality of the LLM's responses at the semantic level. In particular, a semantic response $C$ has probability

$$p(C|q) = \sum_{r \in C} p(r|q). \tag{1}$$

Here $p(r|q)$ is the probability of a single response from the LLM.

However, it is non-trivial to define the equivalent class, and we will discuss the approximation later. To estimate $p(C|q)$, one approach is through semantic similarities between response samples of an LLM for the same question $q$. Let $(r_1, \ldots, r_n)$ be $n$ responses from the same question $q$, and they form $k$ clusters $\tilde{\mathcal{C}}_q = \{\tilde{C}_1, \ldots, \tilde{C}_k\}$ by their semantic similarity. We can use Natural language inference (NLI) systems to predict the relationships (e.g., entailment and contradiction) between responses and derive their similarity.

We assume that each cluster cluster $\tilde{C}$ is from a different semantic class $C$, then $p(C|q)$ can be approximated by

$$p(C|q) \approx \frac{|\tilde{C}|}{n}. \tag{2}$$

From the cluster probabilities, the uncertainty of the LLM on the question $q$ is estimated as the entropy of the empirical distribution over clusters (Kuhn et al., 2022), and the confidence of a response $r_i \in C$ is estimated as $|\tilde{C}|/n$ (assuming similarity values are binary) (Lin et al., 2024).

Now, we depart from the setup of semantic classes and consider the correctness of responses. Let $C^*$ be the correct semantic answer to question $q$. Without knowing which responses are correct answers, a common assumption is that the model's confidence reflects the correctness, that is, the model's confidence in a semantic response is approximately the probability of correctness, then

$$p(\tilde{C}_{k'} \subseteq C^*) \approx \frac{|\tilde{C}_{k'}|}{n}. \tag{3}$$

It says that the more certain the model is about a semantic response, the more likely the response is correct. Conversely, a wide variation in the LLM's responses indicates low confidence in all responses $r_i$ and low accuracy. This pattern is also found in previous studies (Kuhn et al., 2022; Lin et al., 2024).

While we agree that a positive correlation exists between the LLM's confidence and the probability of correctness, we do not believe that they are equal, as shown in equation 3. Therefore, we need further calibration to reflect the probability of correctness.

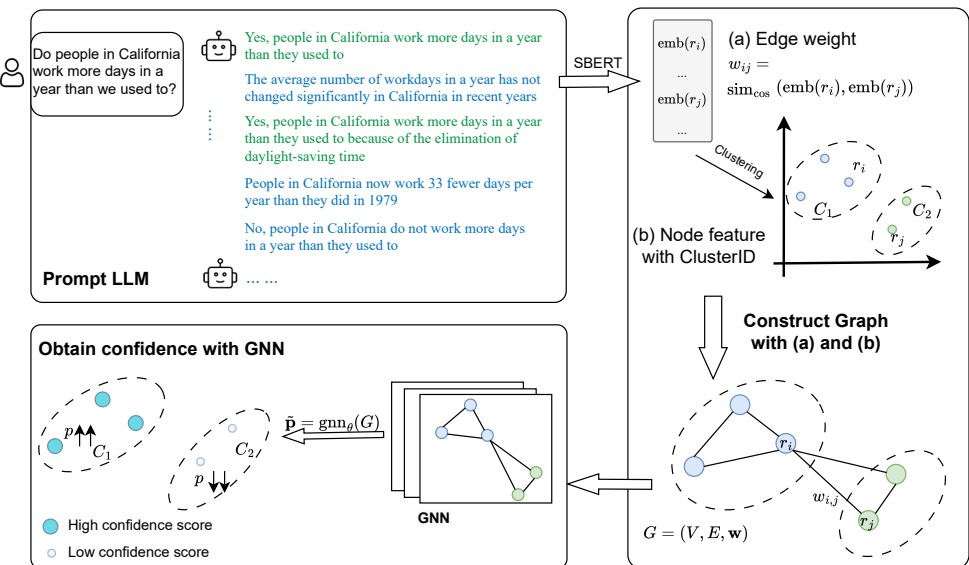

Figure 1: **The overall framework of our confidence calibration model**. Given an input question, our approach first generates multiple responses from the LLM and constructs a similarity-weighted graph based on these responses. This graph serves as the input for the GNN model, which calibrates the confidence of the LLM responses. In the weighted graph, the edge weight $w_{ij}$ is defined as $\text{sim}_{\cos}(\text{emb}(r_i), \text{emb}(r_j))$, where $i, j = 1, \ldots, n$. A higher weight indicates greater similarity between the responses. For the node features, we use the clusterID, refers to the cluster number assigned to each response.

### 3.1 CONFIDENCE CALIBRATION AS GRAPH LEARNING PROBLEM

Now, we set a supervised learning problem and train a model to calibrate the confidence of the correctness of responses. We first consider the correctness labels of the LLM's responses. In the supervised setting, we have a correct answer $r^*$ to the question $q$. Then $r^*$ to assign correctness labels to sampled responses $\{r_1, r_2, ..., r_n\}$ for the same question $q$. In our work, we use the ROUGE similarity. Specifically, we compute the ROUGE similarity $\text{sim}_R(r_i, r^*)$ between a sampled response and the correct answer to decide the correctness label.

$$y_i = \mathbb{1}[\text{sim}_R(a, r_i) \geq \tau], \; i = 1, \ldots, n. \tag{4}$$

Here $\mathbb{1}[\cdot]$ is one if the condition is true or 0 otherwise. The ROUGE metric is reasonably accurate in measuring semantic similarity between short sentences (Lin & Och, 2004). We follow the previous work, and set $\tau = 0.3$ (Kuhn et al., 2022).

In the second method, we utilize the LLM to generate correctness labels. Specifically, we provide the question $q$ and the standard answer $a$ as the context, then ask whether the response $r_i$ answers the question $q$. The response from the LLM is then used as the label for $r_i$. We denote the procedure as

$$y_i = \text{llm}_y(q, a, r_i) \tag{5}$$

We provide the prompt for labeling in the Appendix F. We then consider the input to the calibration model. We form a similar graph $G$ over responses to encode information about their consistency. The graph contains the clustering structure of responses and likely further useful information to predict the correctness of responses. The graph $G = (V, E, \mathbf{w})$ is a fully connected graph, with the node set $V$ consisting of $n$ responses and the edge weight $w_{ij}$ being the similarity between the pair of responses $(r_i, r_j)$. We compute the similarity from the two responses' embeddings. In particular,

we first use the Sentence-BERT model (Reimers & Gurevych, 2019) to compute the two responses' vector representations and then compute the cosine similarity

$$w_{ij} = \text{sim}_{\cos}(\text{emb}(r_i), \text{emb}(r_j)), \ \ i, j = 1, \dots, n. \tag{6}$$

Here, $\text{emb}(\cdot)$ represents the embedding function.

Then, we treat the problem as a node classification problem (Xiao et al., 2022). In particular, we run a Graph Neural Network (GNN) $\text{gnn}(\cdot)$ to predict the probability of each response being correct

$$\tilde{\mathbf{p}} = \text{gnn}_\theta(G). \tag{7}$$

Here $\tilde{\mathbf{p}} \in [0, 1]^n$ contains the probabilities for $n$ responses being correct.

To provide clustering information to the GNN, we first run the K-means clustering algorithm on the responses' embeddings and assign cluster IDs from 0 to $K - 1$ based on the order from largest to smallest (ties are randomly broken). Then, we feed each response's cluster membership as a one-hot feature input to the GNN. Therefore, the GNN's predictions are purely based on the relationships between responses in semantic space. We choose NOT to feed in the embedding vectors of responses to avoid the GNN's dependency on textual information. This helps the GNN to generalize to questions from different domains. The overall framework is shown in Fig 1.

The main purpose of the learning model is to calibrate $\tilde{\mathbf{p}}$. One approach is to minimize the cross-entropy loss of $\tilde{\mathbf{p}}$ against correctness labels. The loss computed from the question $q$ is

$$\ell_q = -\sum_{i=1}^n y_i \log \tilde{p}_i + (1 - y_i) \log(1 - \tilde{p}_i) \tag{8}$$

Note that the loss is consistent marginally since the loss is minimized when $\tilde{p}_i = p(y_i | G)$. An alternative approach is to minimize the squared error $(y_i - \tilde{p}_i)$, from which we get similar performances, so we choose the cross-entropy loss. A further consideration is to explicitly consider the similarity between $\tilde{p}_i$ and $\tilde{p}_j$ given the response similarity $w_{ij}$. We leave such exploration to the future.

### 3.2 IMPROVE THE ESTIMATION THROUGH MULTIPLE PROMPTS

It is well known that the syntactic form of a question influences responses and introduces additional variance. To reduce this variance and evaluate the LLM's semantic consistency, we analyze the LLM's responses to multiple prompts derived from the same question. These responses are treated as answers to the same semantic question. We then apply the same method as before to predict the correctness of each response.

In particular, we rephrase the original question $q$ into $k$ different forms $\{q_1, ..., q_k\}$ while maintaining the original sentence's semantic meaning. We employ a multiple rephrased questions strategy for answer sampling. Specifically, we prompt the GPT-4 to give $k$ different but with the same meaning rephrased questions for the given question $q$. Then, we sample $n/k$ responses from the LLM for each rephrased question and still get a total of $n$ responses, from which the confidence calibration is the same as we have described above. For questions about which the LLM is less certain, the model is more likely to produce diverse responses. In this scenario, confidence calibration is more accurate because the model's uncertainty becomes more apparent.

## 4 EXPERIMENTS

The goal of this section is to compare our proposed framework with baseline methods in terms of confidence calibration. All experiments are conducted on NVIDIA A100 GPUs with 80GB of memory. The supplementary materials and Appendix provide the code for our model, more experiment details in Appendix A, and prompting strategy and Appendix F.

### 4.1 DATASET AND EXPERIMENT SETUP

**Dataset**: We conduct experiments on two public benchmark datasets: (1) CoQA (Reddy et al., 2019), an open-book conversational question answering (QA) task; (2) TriviaQA (Joshi et al., 2017), a commonsense QA task. and (3) TruthfulQA (Lin et al., 2022a), a comparably more challenging

dataset for factual QA tasks. We use the first 4k as training data and 1582 for validation and testing, both of equal size. For TruthfulQA, we use the 7:1:2 split for training, validation, and testing. We provide more details about training in the Appendix A.

**Baselines**: We compare our methods with the following baselines. **Length-normalized sequence likelihoods (Seq. likelihood)** (Malinin & Gales, 2021; Kuhn et al., 2022) is a standard measure for confidence. This method calculates the likelihood of each sequence and normalizes it by the length of the sequence to provide a fair comparison between different lengths of sequences. **Platt scaling** (Platt, 1999), a variant of the sequence likelihood baseline, applies Platt scaling to the raw likelihoods. **GraphSpectral** (Chen et al., 2024) uses the graph theory to estimate the confidence. Then we also include post-hoc uncertainty calibration, **GraphSpectral+Iso** and **GraphSpectral+Platt** into the baseline methods. **Self-check GPT**(Manakul et al., 2023) checked the consistency between responses querying the LLM. **Verbalized Uncertainty** (Lin et al., 2022b; Tian et al., 2023; Xiong et al., 2024) generates verbal statements about the model's confidence in its predictions. Verbalized Qual maps the confidence percent (Verbalized %) into numerical values. **APRICOT** (Ulmer et al., 2024), a supervised method, fine-tunes the Deberta language model to predict confidence scores for LLM outputs. Furthermore, we also include the baseline of applying two post-hoc uncertainty calibration methods, **APRICOT+Iso** and **APRICOT+Platt**, to adjust the confidence scores obtained by Apricot. We performed all the baseline experiments utilizing the open-source codebase and used the default parameters.

**Graph construction:** For each question, we prompt the LLM to generate 30 answers. Each generated answer is then processed using the SentenceBert model Reimers & Gurevych (2019) to obtain the answer's high-dimensional embeddings. To quantify the semantic similarity between the answers, we compute the cosine similarity between every pair of answer embeddings. These similarity scores are then utilized as edge weights in our similarity graph, where each node represents an individual answer, and the edges signify the degree of semantic relation between them.

**Model hyper-parameters**: To ensure our model can capture complex and abstract features at each layer, our model comprises three Graph Convolutional Network (GCN) layers, with embedding dimensions of 256, 512, and 1024 for the first, second, and third layers, respectively. The initial learning rate was set to $10^{-4}$. If the validation loss did not show improvement over ten consecutive epochs, the learning rate was reduced by a factor of 0.9. The optimization was performed using the Adam optimizer, configured with hyperparameters $\beta_1 = 0.9$ and $\beta_2 = 0.98$. The batch size was 16.

**LLMs**: We assess our confidence calibration method on two LLMs with excellent performance: `Llama3-8B` (Llama3)(Meta, 2024), and `Vicuna-7b-v1.5` (Vicuna) (Zheng et al., 2024).

**Labeling the data**: To obtain the correctness label for CoQA and TriviaQA datasets, we followed previous work (Kuhn et al., 2022) and used the Rougel-L metric for labeling. For the TruthfulQA dataset, given its focus on factual correctness and longer answers, we employed GPT4 Liu et al. (2023) to generate the labels.

**Evaluation metrics**: The evaluation metrics include Expectation Calibration Error (ECE), Brier Score, and AUROC. Specifically, (1) **ECE** quantifies the consistency between the prediction error and the uncertainty of the prediction. An ideal calibration curve should exhibit a lower ECE. It measures the consistency between the prediction error and the confidence of the prediction. Specifically, the confidence interval is grouped into fixed bins, and the average of the difference between the confidence and error in each bin is compared. Formally, ECE is calculated as $ECE = \sum_{b=1}^{B} \frac{n_b}{N} |acc(b) - conf(b)|$, where $n_b$ is the number of predictions in bin $b$, $N$ is the total number of data points and $acc(b)$ and $conf(b)$ are the accuracy and confidence of bin b, respectively. (2) **Brier Score** (Brier, 1950), which is the mean squared difference between predicted probabilities and the actual binary results. Lower Brier Scores indicate better performance. (3) **AUROC** to indicate the models' discriminatory ability.

## 4.2 EXPERIMENT RESULTS

For the Llama3 model, the confidence calibration performance on TriviaQA, CoQA, and TruthfulQA are shown in Table 1. For the TriviaQA dataset, it can be observed that the likelihood-based method performs poorly on the calibration error (ECE and Brier Score) and AUROC due to unreliable model prediction probability (Zhang et al., 2024). Platt scaling improves the ECE post-calibration and en-

| Method | TriviaQA | | | CoQA | | | TruthfulQA | | |
|---|---|---|---|---|---|---|---|---|---|
| | Brier↓ | AUROC↑ | ECE↓ | Brier↓ | AUROC↑ | ECE↓ | Brier↓ | AUROC↑ | ECE↓ |
| GraphSpectral (GS) | 0.22 | 0.80 | 0.056 | 0.19 | 0.76 | 0.091 | 0.33 | 0.63 | 0.312 |
| GS + Iso | 0.16 | 0.80 | 0.072 | 0.16 | 0.76 | 0.054 | 0.19 | 0.63 | 0.091 |
| GS + Platt | 0.16 | 0.79 | 0.048 | 0.16 | 0.74 | 0.041 | 0.22 | 0.63 | 0.151 |
| Self-check GPT | 0.33 | 0.65 | 0.187 | 0.21 | 0.63 | 0.178 | 0.36 | 0.56 | 0.353 |
| Seq. likelihood | 0.54 | 0.53 | 0.22 | 0.38 | 0.47 | 0.17 | 0.46 | 0.42 | 0.03 |
| Platt | 0.28 | 0.59 | 0.05 | 0.26 | 0.47 | 0.09 | 0.27 | 0.57 | 0.03 |
| Verbalized Qual | 0.32 | 0.62 | 0.14 | 0.30 | 0.68 | 0.16 | 0.32 | 0.62 | 0.14 |
| Verbalized % | 0.25 | 0.67 | 0.033 | 0.42 | 0.66 | 0.21 | 0.54 | 0.57 | 0.33 |
| APRICOT | **0.14** | 0.76 | 0.074 | 0.17 | 0.78 | 0.132 | 0.20 | 0.69 | 0.173 |
| APRICOT + Iso | 0.18 | 0.76 | 0.073 | 0.17 | 0.78 | 0.097 | 0.20 | 0.68 | 0.112 |
| APRICOT + Platt | 0.17 | 0.76 | 0.039 | 0.17 | 0.78 | 0.069 | 0.23 | 0.69 | 0.131 |
| Ours | **0.14** | **0.82** | **0.016** | 0.12 | 0.77 | 0.016 | 0.18 | **0.70** | **0.026** |
| Ours (Multi prompts) | **0.14** | **0.82** | **0.016** | 0.12 | **0.78** | **0.015** | 0.17 | **0.70** | **0.026** |

Table 1: Comparison of confidence calibration performance on TriviaQA, CoQA and TruthfulQA dataset for Llama3

hances the model's discriminative ability, resulting in higher AUROC results. However, this method cannot capture the semantic equivalence among answers, leading to sub-optimal performance. The Verbalized and Verbalized Qual prompts LLM to output confidence for their answers, improving AUROC by $3 - 5\%$ compared with the likelihood baseline. However, it faces the overconfidence issue; thus, the calibration errors are still high. The GraphSpectral method can produce good confidence estimations, but its calibration performance is poor. Even with the addition of techniques such as Isotonic Calibration or Platt Scaling, this issue can only be partially mitigated. The auxiliary DeBERTa method combines the LLM outputs, Chain-of-Thoughts (CoT) outputs, and verbalized confidence to fine-tune the DeBERTa model for predicting confidence. Our method captures the prediction confidence based on the graph structure of LLM's responses in semantic space and achieves better ECE results. The ECE is reduced from $0.07$ to $0.016$ and improves the AUROC from $0.76$ to $0.82$ compared with the baseline calibration methods. The experiment results on TruthfulQA and CoQA for the Llama3 model are shown in Table 1. These results show a similar trend, with our model achieving superior performance in confidence calibration compared to the baseline methods.

| Method | TriviaQA | | | CoQA | | | TruthfulQA | | |
|---|---|---|---|---|---|---|---|---|---|
| | Brier↓ | AUROC↑ | ECE↓ | Brier↓ | AUROC↑ | ECE↓ | Brier↓ | AUROC↑ | ECE↓ |
| GraphSpectral (GS) | 0.19 | 0.79 | 0.112 | 0.27 | 0.69 | 0.202 | 0.286 | 0.64 | 0.226 |
| GS + Iso | 0.19 | 0.79 | 0.059 | 0.24 | 069 | 0.037 | 0.297 | 0.64 | 0.092 |
| GS + Platt | 0.17 | 0.79 | 0.067 | 0.22 | 0.69 | 0.055 | 0.307 | 0.64 | 0.183 |
| Self-check GPT | 0.35 | 0.63 | 0.18 | 0.22 | 0.64 | 0.192 | 0.28 | 0.55 | 0.308 |
| Verbalized Qual | 0.38 | 0.62 | 0.02 | 0.45 | 0.48 | **0.00** | 0.47 | 0.48 | **0.01** |
| Verbalized % | 0.39 | 0.52 | 0.38 | 0.49 | 0.53 | 0.32 | 0.58 | 0.56 | 0.38 |
| Seq. likelihood | 0.47 | 0.58 | 0.42 | 0.30 | 0.68 | 0.16 | 0.32 | 0.50 | 0.20 |
| Platt | 0.34 | 0.58 | 0.25 | 0.30 | 0.68 | 0.16 | 0.28 | 0.58 | 0.18 |
| APRICOT | 0.18 | 0.78 | 0.068 | 0.19 | 0.74 | 0.073 | **0.19** | 0.76 | 0.11 |
| APRICOT + Iso | 0.17 | 0.78 | 0.049 | 0.19 | 0.74 | 0.06 | **0.19** | 0.76 | 0.09 |
| APRICOT + Platt | 0.17 | 0.78 | 0.052 | 0.19 | 0.74 | 0.049 | 0.20 | 0.76 | 0.08 |
| Ours | 0.17 | **0.81** | 0.020 | 0.18 | **0.75** | 0.02 | 0.20 | **0.77** | **0.05** |
| Ours (Multi prompts) | **0.16** | **0.81** | 0.018 | **0.16** | 0.76 | 0.016 | 0.20 | 0.76 | **0.05** |

Table 2: Comparison of confidence calibration performance on TriviaQA, CoQA and TruthfulQA dataset for Vicuna

Furthermore, we also compare the confidence calibration performance for the Vicuna model on the TriviaQA, CoQA, and TruthfulQA datasets. The results are summarized in Table 2. Our model consistently improves the calibration error compared to the baseline methods.

To understand the improvements of our model, we present the reliability diagrams for the baseline methods applied to the Vicuna model on TriviaQA. The reliability diagram is created by discretizing

the confidence value into 10 bins and then computing the average accuracy for samples in each bin. The ideal calibration curve should align with the diagonal line, indicating that the confidence value can match the probability of correctness. The reliability diagram is shown in Fig. 2. (We also show other reliability diagrams for the different methods for Llamas on TriviaQA and CoQA in the Appendix E). The figure presents the reliability diagrams for different methods, each utilizing 10 bins. In these diagrams, both the color intensity and the percentage numbers within each bar represent the proportion of total responses that fall into each respective bin. Specifically, larger proportions are depicted with colors closer to purple, while the height of each bar indicates the ratio of correct predictions within that bin. An ideal reliability diagram should exhibit a wide distribution of responses across multiple bins, demonstrating the model's strong ability to differentiate between varying confidence levels in its predictions. Additionally, the heights of the bars should align closely with the

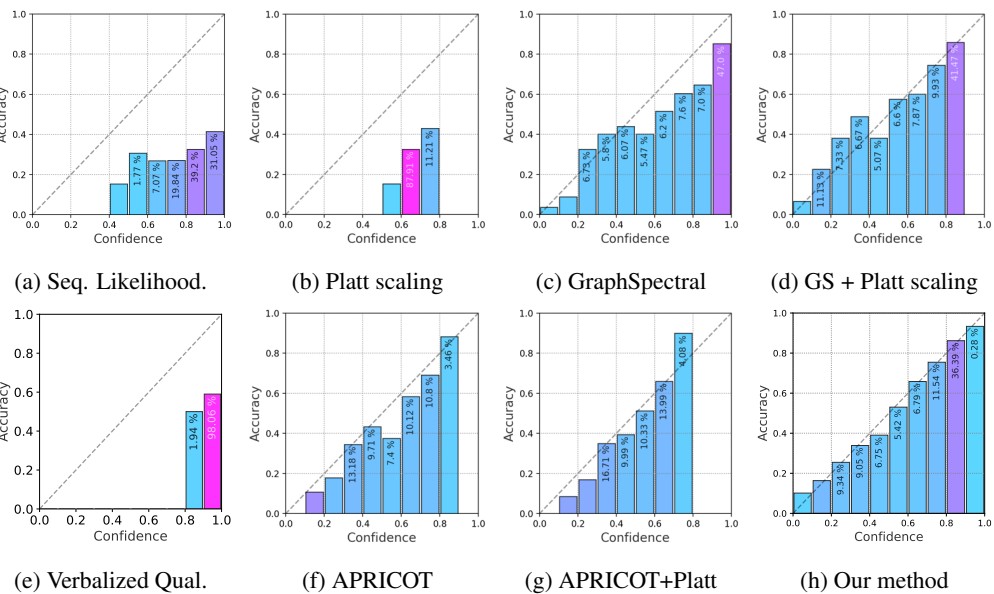

Figure 2: Reliability diagrams for different methods using 10 bins each for Vicuna on TriviaQA. The color, as well as the percentage number within each bar, indicates the proportion of total responses contained in each bin. Larger values are represented by colors closer to purple, and the height indicates the ratio of correct ones. We prefer a wide spread of responses in different bins (strong ability to differentiate responses) and bin heights along the diagonal line (accurate calibration). Our model outperforms others with a broader bin spread and better alignment with the diagonal for calibration accuracy.

diagonal line, which represents perfect calibration—where the predicted confidence matches the empirical accuracy. It can be observed that the likelihood-based confidence methods exhibit significant overconfidence, with curves below the diagonal, indicating many samples have high confidence but low accuracy. This results in poor ECE performance. Although the Platting scaling calibration method enhances the ECE performance, it still has poor AUROC. The Auxiliary DeBERTa (APRICOT) method, which integrates LLM outputs, Chain-of-Thought (CoT) outputs, and verbalized confidence to train an auxiliary DeBERTa model, enhances the AUROC. However, it still experiences some overconfidence issues, potentially caused by the inherent overconfidence in the input verbalized confidence scores. The baseline methods' reliability diagrams revealed that this method frequently assigned high confidence scores to incorrect predictions, deviating markedly from the ideal calibration represented by the diagonal line. For example, the verbalized method's predictions in the highest confidence bins (80-90%) were significantly below the corresponding empirical accuracy, indicating a tendency to overestimate the certainty of its outputs. In contrast, our framework achieves a broad spread of responses across the bins, showing good differentiation capabilities; at the same time, the bar heights closely follow the diagonal line, indicating better calibration.

## 4.3 OUT OF DOMAIN EVALUATION

Domain shift poses significant challenges for deploying machine learning models in real-world scenarios where data variability is expected. To comprehensively assess the robustness and generalization capabilities of our proposed model compared to baseline methods, we conducted a series of out-of-domain (OOD) evaluations.

**Experiment setup:** We evaluate the confidence calibration of different approaches under out-of-domain settings. We have two experiment configurations: **out-of-domain dataset OODD**, and **out-of-domain LLMs (OODL)**. For OODD, we train the confidence calibration model on TriviaQA from Llama3 responses and test it on CoQA Llama3 and TruthfulQA Llama3 answers. For OODL, we use the same training data from Llama3 but test the Vicuna model's responses on the TriviaQA and CoQA datasets. We compare our model with the Apricot and GraphSpectral (with Platt scaling) methods.

**Results and Analysis:** Table. 3 shows the OOD performance of the baseline methods. The OOD experiment results revealed that our model maintained a high level of performance across tested domains. Specifically, the model demonstrated consistent calibration, as evidenced by low ECE values and strong discriminative ability, reflected in high AUROC scores on in-domain and OOD datasets. For example, while the model achieved an ECE of 0.016 and an AUROC of 0.82 on TriviaQA (in-domain), it maintained an ECE of 0.077 and an AUROC of 0.77 on CoQA. Furthermore, the Brier scores across domains remained within acceptable ranges, demonstrating reliable probabilistic predictions even when faced with unfamiliar data distributions. The relatively small increase in ECE and a slight decrease in AUROC for OOD datasets suggest that while there is some degradation in performance, the model retains substantial robustness and accuracy. This is primarily because similarity graph patterns are highly invariant to the data distribution. Specifically, our model employs the consistency graph and the clustering feature that does not alter with data distribution shifts, enabling it to maintain stable performance across different datasets.

| Dataset | Method | Brier | AUROC | ECE |
|---|---|---|---|---|
| llama3 CoQA | GraphSpectral(w platt scaling) | 0.17 | 0.72 | 0.095 |
| | Apricot | 0.24 | 0.59 | 0.154 |
| | Ours | **0.13** | **0.77** | **0.077** |
| llama3 TruthfulQA | GraphSpectral(w platt scaling) | 0.32 | 0.63 | 0.324 |
| | Apricot | 0.25 | 0.54 | 0.197 |
| | Ours | **0.23** | **0.66** | **0.16** |
| Vicuna TriviaQA | GraphSpectral(w platt scaling) | 0.24 | 0.53 | 0.07 |
| | Apricot | 0.19 | 0.76 | 0.13 |
| | Ours | **0.17** | **0.81** | **0.07** |
| Vicuna CoQA | GraphSpectral(w platt scaling) | 0.35 | 0.55 | 0.26 |
| | Apricot | 0.24 | 0.59 | **0.08** |
| | Ours | **0.22** | **0.73** | 0.10 |

Table 3: OOD evaluation for models trained on the TriviaQA from Llama3 responses and tested out-of-domain datasets

In contrast, Apricot typically relies on specific dataset features, which leads to poor performance in OOD scenarios. Furthermore, calibration methods like the Platt scaling can improve the confidence calibration in-domain, but their calibration effectiveness remains limited under domain shift scenarios. This is because this calibration technique mainly adjusts the output probabilities but does not fundamentally address the biases introduced by feature representation changes across distributions.

## 4.4 SENSITIVITY ANALYSIS

In this subsection, we conducted several sensitivity analyses of our model.

**Number of training samples** We conducted experiments to examine the relationship between performance and the amount of training data. Specifically, we tested our model performance on the Llama3 TriviaQA dataset and varied the training size from 100 to 4000. The results are displayed

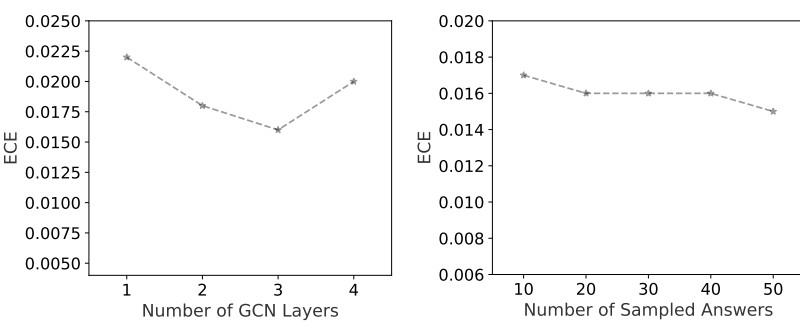

(a) Sensitivity of the number of GCN layers    (b) Sensitivity of the number of node

Figure 3: Sensitivity analysis of our model

in Table 4. We observed that the model's performance does not drop significantly with the reduced training data. These experimental results indicate that the model performs well with limited data availability, demonstrating its applicability in real-world scenarios where only smaller datasets are available. We also tested the baseline performance, the results are shown in Appendix E.

Table 4: Performance under varying Training Sample Sizes

| # of Training Samples | ECE | AUROC | Brier |
|---|---|---|---|
| 100 | 0.095 | 0.770 | 0.201 |
| 300 | 0.062 | 0.786 | 0.187 |
| 500 | 0.049 | 0.792 | 0.181 |
| 1000 | 0.037 | 0.799 | 0.177 |
| 4000 | 0.016 | 0.820 | 0.172 |

**Hyperparameter sensitivity** We conduct the sensitivity analysis of our model's calibration error performance concerning two key configurations: the number of sampled answers used to construct the graph and the number of Graph Convolutional Network (GCN) layers in the GNN model. The results are displayed in Fig. 3. The experiments are conducted using the Llama3 model on the TriviaQA dataset. For Fig. 3 (a) experiments, we varied the number of sampled answers from 10 to 50 while keeping other configurations and hyperparameters fixed, as described in the experimental setup. We observe that increasing the number of sampled answers slightly improves performance, which then stabilizes. In Fig. 3(b), the sensitivity to the number of GCN layers indicates that our model remains stable with 1 to 4 layers, with the best performance observed at 3 layers.

## 5 CONCLUSION AND FUTURE WORK

In summary, in this work, we proposed one effective strategy of confidence calibration by combining the LLM's self-consistency with labeled data and training an auxiliary GNN model to estimate the correctness of its responses to questions. Experiments demonstrate that the proposed approach improves confidence calibration significantly across several datasets compared to baseline methods. Our calibration model enhances the reliability of LLMs by evaluating response accuracy, enabling them to abstain from uncertain queries and empowering users to determine trust levels, thereby promoting responsible deployment in society. However, there are instances where an LLM might be highly confident in an incorrect semantic response, resulting in a consistency graph similar to that of a correct answer. In such cases, our calibration model may not provide an accurate confidence estimation. Unfortunately, without a model stronger than the LLM itself, there is no straightforward solution to this problem. We hope that advancements in LLMs will help mitigate this issue. In future work, we aim to extend the framework to incorporate the data uncertainty coming from ambiguous questions and also explore the multi-step confidence calibration in the chain-of-thought framework.

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

## A HYPERPARAMETERS AND MODEL CONFIGURATIONS

**Model hyper-parameters**:

Our model used three GCN layers; typically, the embedding dimension was $256, 512, and 1024$ for three GCN layers. For the training process, we used the binary cross-entropy loss with a decaying learning rate that reduced the learning rate by $0.9$ if the validation loss did not improve 10 epochs (with an initial learning rate of $10^-4$ and a minimum learning rate of $10^{-7}$). The optimizer was Adam with $\beta_1 = 0.9$ and $\beta_2 = 0.98$. The batch size was 32. For the rephrased prompts, we set $k = 3, n = 30$, so for each rephrased question, we sampled ten answers. While calculating the ECE, we divide the confidence into $B = 10$ bins.

**Evaluation Setup:**

For each question, we evaluate the confidence prediction corresponding to the most likely answers from the LLM response. The setup is consistent with the baseline methods.

**Graph construction:**

For each question, we prompt the LLM to give 30 answers, and the temperature for LLM is set to be 0.6. For each answer, the SentenceBert model Reimers & Gurevych (2019) is used to get each answer's embedding. The cosine similarity between each answer's embedding is taken as the edge weight of the graph. We apply the K-Means clustering method to cluster similar semantic responses. The maximum cluster number is set as 3.

## B   COMPUTATIONAL COST

We performed all experiments on NVIDIA A100 GPUs with 80GB of memory. Generating 30 responses using the Llama3 and Vicuna models for 6000 questions from CoQA and TriviaQA data required up to 4 hours, with an average of approximately 2 seconds per question. The CoQA dataset demanded more processing time due to the longer contextual information in the input. The time can be shortened by parallel sampling.

## C   ADDITIONAL CASES

To better understand our method intuitively, we have collected a few examples to show the difference between our algorithm and APRICOT.

To summarize our observation here:

1. Multiple responses to the same question does reveal the LLM's confidence in its answers. 2. The LLM's self-evaluation of confidence is often much higher than it should be – the LLM is overconfident about its responses. 3. The chain-of-thought responses used by ApriCoT add some information to make each answer more complete and reasonable in the spirit of 1, but it mainly adds the information within one response, not as much information as the multiple responses used by ours.

**Example 1:**

Question: Who plays Captain Jack Sparrow's father Edward Teague in the Pirates of the Caribbean films?

True answer:: Keith Richards

LLM response: David Schofield

More responses from the LLM: Martin Klebba. Keith Richards, Geoffrey Rush, Martin Klebba. Keith Richards. Martin Klebba. David Schofield. (only list 7 responses here to save space)

GCC-estimated confidence: 0.23

CoT response: David Schofield,

Self-evaluation: 80

ApriCoT-estimated confidence: 0.79

**Example 2:**

Question: In which film will you find the Rodger Young?

True answer:: Starship Troopers

LLM response: The Bridge on the River Kwai.

More responses from the LLM: The Greatest Story Ever Told. The Best Years of Our Lives. The Bridge on the River Kwai. The Best Years of Our Lives (1946). 1949's Battleground. The Best Years of Our Lives.

GCC-estimated confidence: 0.22

CoT response: All the President's Men.

Self-evaluation: 95

ApriCoT-estimated confidence: 0.81

**Example 3:**

Question: BS is the international car registration of which country?

True answer:: Bahamas.

LLM response: Germany.

More responses from the LLM: Bahamas. Bahrain. Bangladesh. Bahamas. Belgium. Bahamas. Germany. Bhutan. Belgium.

GCC-estimated confidence: 0.34

CoT response: Belgium

Self-evaluation: 98

ApriCoT-estimated confidence: 0.61

## D    ADDITIONAL VISUALIZATIONS

Besides the cases we show in the previous section. Here, we present several case examples and visualize the response patterns. We performed dimension reduction of LLM's responses to different questions and then plotted their embeddings to the 2-dimensional space. Fig 4 shows the responses generated by Llama3 as an example. From the figure, we observe that answers with higher confidence levels tend to cluster closely together, indicating consistency and reliability in these responses. In contrast, answers with lower confidence levels exhibit greater diversity, reflecting a broader range of possibilities. This behavior aligns well with our initial assumption, demonstrating that higher confidence responses are more consistent, while lower confidence responses capture a wider variety of potential answers.

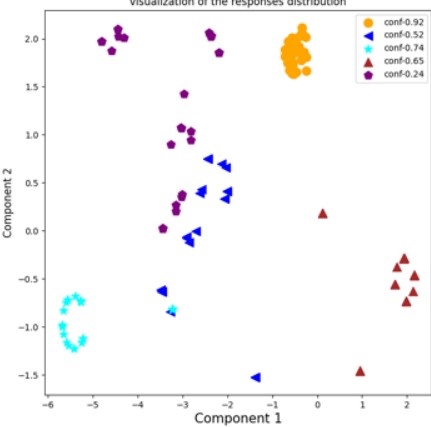

Figure 4: Visualization of the generated response patterns

## E    ADDITIONAL RESULTS

**Additional reliability plots** We showed all reliability diagrams for Llama3 for TriviaQA in Fig. 5 and CoQA dataset in Fig. 6. To summarize the trends, we observe that Platt scaling narrows the range to the middle value. Verbalized uncertainty cannot generate a wider range of confidence values. GraphSpecral with Platt tends to generate a wider range of confidence values, but the bias can not be improved across all cases, resulting in the bar height not following the diagonal line closely. Our model can predict a wider range of confidence values and achieve better calibration in all settings, with the auxiliary consistency graph and clustering features contributing to improved calibration overall.

**Additional baseline results** In Table 5, we showed the performance of the baseline method under varying training sizes. As the number of training data decreases, the ece will drop from 0.096 to 0.165.

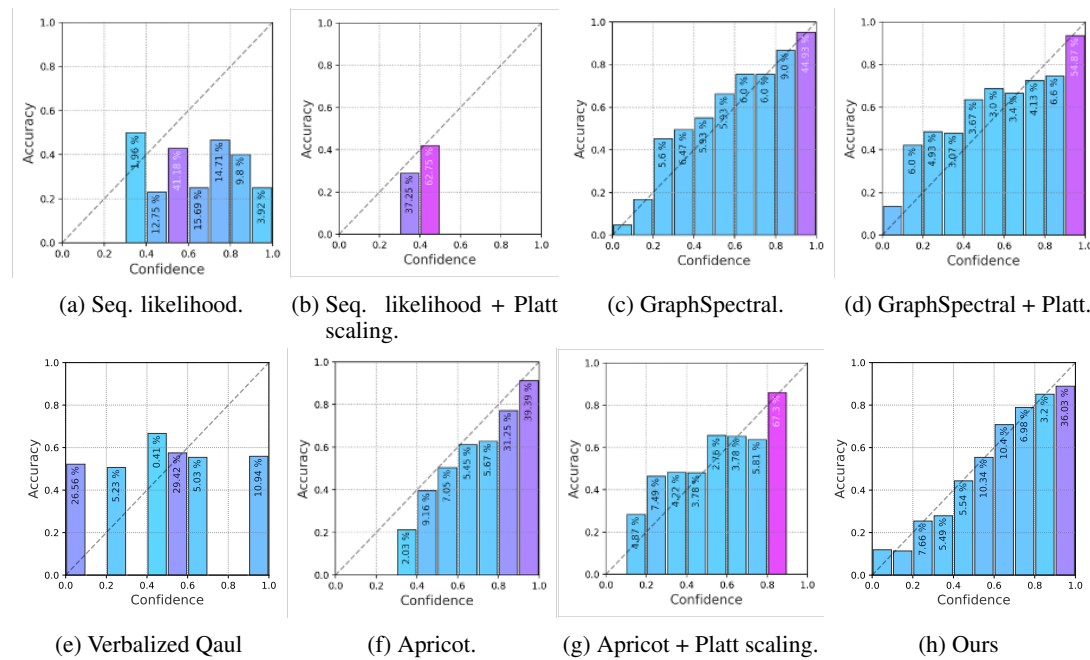

Figure 5: Reliability diagrams for different methods using 10 bins each for TriviaQA from Llama3 model responses. The color and the percentage number within each bar indicate the ratio of responses contained in each bin. Larger values are represented by colors closer to purple.

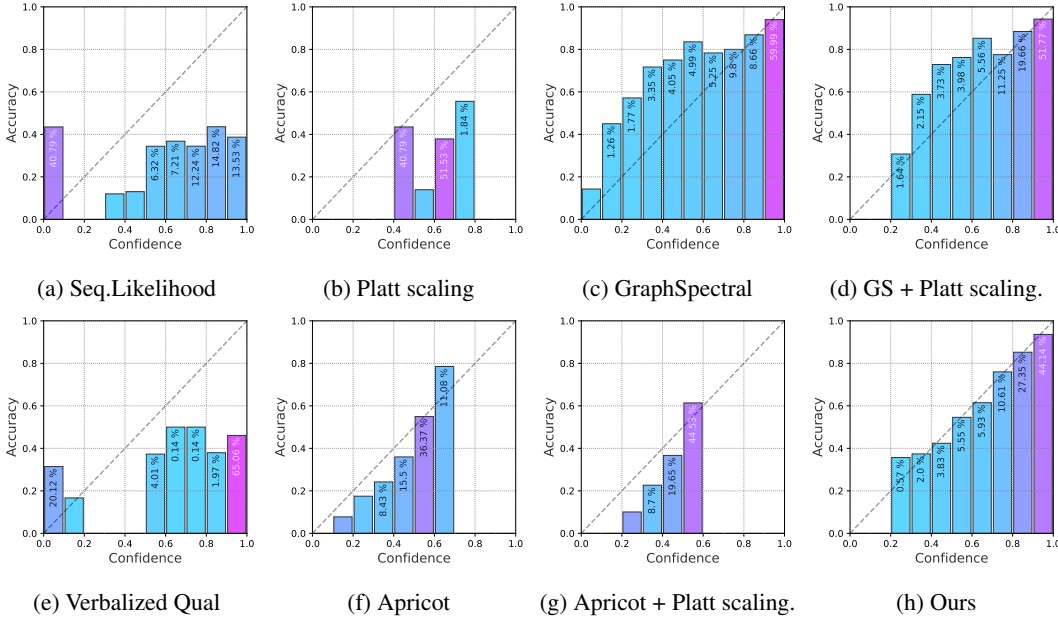

Figure 6: Reliability diagrams for different methods using 10 bins each for CoQA from Llama3 model responses. The color and the percentage number within each bar indicate the ratio of responses contained in each bin. Larger values are represented by colors closer to purple.

## F PROMPTING STRATEGY

Here, we showed the prompts to generate the rephrasing questions.

Table 5: Performance under varying Training Sample Sizes for the baseline methods(Apricot)

| # of Training Samples | ECE | AUROC | Brier |
|---|---|---|---|
| 100 | 0.165 | 0.611 | 0.229 |
| 300 | 0.133 | 0.634 | 0.211 |
| 500 | 0.112 | 0.695 | 0.204 |
| 1000 | 0.105 | 0.722 | 0.192 |
| 4000 | 0.096 | 0.743 | 0.187 |

**Prompts for rephrasing questions**

You are a helpful assistant. I have a question that I would like to see it rephrased in multiple ways. Please take the original question and generate several rephrased versions while maintaining the same meaning, and the question can only have one direct answer. Here is the original question: . . . . Please provide four distinct rephrases of the question.

The prompts for labeling:

**Prompts for labeling**

You will be provided with a question, a reference answer, and a student's answer. Please evaluate the student's answer based on the reference answer and provide your score for the student's answer in the format: "Score: ". Assign a score of 0 for incorrect and 1 for correct. For example, "Score: 0" or "Score: 1". Do not include any additional information. Question: {. . . } Student answer: {. . . } Reference answer: {. . . } Now, please enter your score. Score:

