# OpenReview forum: "Graph-based Confidence Calibration for Large Language Models"
_ICLR.cc/2025/Conference — ICLR 2025 Conference Withdrawn Submission_

### Official Review · Reviewer_RPHJ · 2024-11-01

**Soundness:** 2
**Presentation:** 2
**Contribution:** 2
**Rating:** 5
**Confidence:** 4

**Summary:**

This work proposes a GNN-based approach to calibrate confidence scores of LLM responses. The proposed method is an extension of the Semantic Uncertainty,  a sampling-based method that estimates uncertainty by using the semantic distribution.
The main distinction of the proposed method is that it treats each response as a node and applies GNNs to obtain the logit. Next, cross-entropy loss is introduced to calibrate the node score. Experimental results on short-text datasets CoQA and TriviaQA show that the new method can outperform other baselines and lead to a better generality in OOD settings.

**Strengths:**

1. the paper is easy to follow
2. This work develops a graph-based method to determine the semantic uncertainty, which is interesting
3. better performance in OOD settings

**Weaknesses:**

* **Main concern**:  the scientific contribution of this work is limited. The proposed method is an extended work of the semantic uncertainty with GNNs
* writing and presentation should be improved. Some claims appear too arbitrarily without any proof. Figure 1 is not very informative. (see questions below)
* the proposed method requires labels, but other baselines such as self-checkgpt and semantic uncertainty require no further training
* the use of the ROUGE to evaluate semantic equivalence is problematic since it does not capture semantic meaning well; in the experimental part, only short-answer datasets are used. It is important to include long-form generation dataset

**Questions:**

1. Figure 1 is not very informative, it does not provide the intuition why the proposed method chose to use GNNs
2. In line 163-164, it is supposed to provide proof here: theoretical studies, empirical studies, or examples
3. the use of the ROUGE metric for judging the correctness is problematic. ROUGE does not account for semantic meanings and is sensitive to length and word orders. Therefore, it is not an appropriate metric to evaluate whether two responses are semantically equivalent
4. CoQA and TriviaQA are short-answer datasets. It is useful to test on datasets with long-form answers such as TruthfulQA, which can show the generality of the proposed method
5. in line 63-64, " A response consistent with more answers tends to have a higher likelihood of being correct". A high consistency is an indicator of high confidence rather than high correctness.

---

> ### Author Response · Authors · 2024-11-25
>
> We're grateful for the valuable comments and thank you for your time and expertise.
>
> 1. Thank you for raising this point. We’d like to clarify our contributions in more detail.
>   - Our work addresses an important **research gap in confidence calibration for LLMs**. Existing methods overlook the role of semantic similarity in the calibration process, which limits their ability to generalize effectively across diverse scenarios. To address this, our approach integrates semantic similarity using a graph-based model, enabling more robust and adaptable confidence estimation across various contexts. This generalization capability is further demonstrated through experiments on out-of-domain datasets, where our model consistently outperforms baseline methods.
>     - While related work on semantic uncertainty is related to our work, it operates in a parallel domain without addressing confidence calibration directly. In contrast, our work proposes a new confidence calibration approach for LLMs, filling a critical research gap.
>     - We also want to point out that our work focuses on **confidence calibration**, aiming to align confidence scores with empirical correctness, ensuring the model’s predicted confidence accurately reflects its true correctness. In contrast, **uncertainty estimation** provides a relative measure of uncertainty without requiring alignment to actual correctness.
> - Our experiments demonstrate that the proposed method consistently outperforms state-of-the-art baselines across multiple datasets, including both in-domain and out-of-domain scenarios.
> - In contrast to methods requiring access to internal LLM mechanisms or extensive task-specific fine-tuning, our approach is lightweight compared to baseline methods. This simplicity enhances its applicability to real-world use cases.
> 2. We thank the reviewer for pointing this out. We have carefully revised the writing to improve clarity and coherence throughout the manuscript. Additionally, we have refined Figure 1 to ensure they are more informative.
> 3. The baseline methods mentioned, such as Self-checkGPT, fall under the category of uncertainty estimation. However, they do not calibrate confidence scores based on empirical correctness. To ensure a fair comparison, we extended the Self-checkGPT method by applying additional calibration steps, such as Platt Scaling, which also needs training to align its outputs with correctness-based confidence. This adjustment allows for a more direct comparison between our method and these baselines.
>
> 4. **Regarding the long-form answers** We have included a long-form generation dataset: TruthfulQA dataset. We use GPT4 to create labels to focus on semantic meanings better.
> ### Table 1: Experimental results for TruthfulQA with Llama3 model
> |Method|AUROC|ECE|Brier|
> |----------|----------|----------|----------|
> |GraphSpectral [Lin et al.]|0.63|0.312|0.33|
> |GraphSpectral + iso calibration|0.63|0.091|0.19|
> |GraphSpectral + platt calibration|0.63|0.151|0.22|
> |Self-check GPT [Manakul et al.]|0.56|0.353|0.36|
> |Apricot[Ulmer et al.]|0.69|0.173|0.20|
> |Apricot + iso calibration |0.68|0.112|0.20|
> |Apricot + platt calibration|0.69|0.131|0.23|
> |**Ours**|**0.70**|**0.026**|**0.18**|
>
> ### Table 2: Experimental results for TruthfulQA with Vicuna model
> |Method|AUROC|ECE|Brier|
> |----------|----------|----------|----------|
> |GraphSpectral [Lin et al.]|0.64|0.226|0.286|
> |GraphSpectral + iso calibration|0.64|0.092|0.297|
> |GraphSpectral + platt calibration|0.64|0.183|0.307|
> |Self-check GPT [Manakul et al.]|0.55|0.308|0.28|
> |Apricot[Ulmer et al.]|0.76|0.11|**0.19**|
> |Apricot + iso calibration |0.76|0.09|**0.19**|
> |Apricot + platt calibration|0.76|0.08|0.20|
> |**Ours**|**0.77**|**0.05**|0.20|

---

> > ### Author Response · Authors · 2024-11-25
> >
> > 5. **Regarding ine 63-64**: We would like to point out that our assumption is that higher consistency among LLM’s responses indicates a higher probability of correctness. To validate this, we conducted an empirical evaluation on the TriviaQA dataset. On the TriviaQA dataset, we observed a strong statistical correlation between correctness and consistency with R² = 0.966. This suggests that statistically speaking, responses exhibiting greater consistency are more likely to be correct. We further validated this assumption using the TruthfulQA dataset by analyzing the proportion of highly consistent yet incorrect answers. Results showed that only 13% of all incorrect responses exhibited high consistency.
> > Similarly, this trend was observed in other models, such as GPT-4-o1, on the SimpleQA dataset, where more consistent answers demonstrated higher accuracy (see Figure 2 in [1]). While this confirms the general reliability of our assumption, we acknowledge that there are always outliers where the LLM generates consistent but incorrect responses due to systematic biases or errors. While these cases exist, they are statistically rare, and the strong correlation observed in our study supports the utility of consistency as a proxy for correctness in most scenarios. We will include a detailed discussion in the appendix. Furthermore， this assumption aligns with insights from prior work [1][2], further reinforcing its validity as a practical method for calibration.
> >
> > [1] Wei, Jason, et al. "Measuring short-form factuality in large language models." arXiv preprint arXiv:2411.04368 (2024).
> > [2] Lin, Zhen, Shubhendu Trivedi, and Jimeng Sun. "Generating with Confidence: Uncertainty Quantification for Black-box Large Language Models." Transactions on Machine Learning Research.

---

> > > ### Author Response · Authors · 2024-11-27
> > >
> > > Dear Reviewer RPHJ, thank you once again for your valuable feedback.
> > > We have clarified our contribution and conducted additional experiments, which we believe address your concerns about our work. We kindly ask if you could take a moment to review the response at your convenience. Your feedback is important to us, and we look forward to hearing any further thoughts or questions you may have.
> > >
> > > Kind regards,
> > > Authors

---

> ### Comment · Reviewer_RPHJ · 2024-11-28
> **comments after rebuttal**
>
> Thanks for the responses.
>
> Some of my concerns have been addressed by introducing additional experiments on TruthfulQA.
> Although the improvement is marginal, I would like to raise my score.

---

> > ### Author Response · Authors · 2024-12-02
> >
> > Thank you for your comments and for considering an increase in the score.
> >
> > We appreciate your recognition of the additional experiments on TruthfulQA. We would like to note the strength of our method lies in the calibration of the confidence.  Specifically, our method achieves a **74% reduction** in Expected calibration error (ECE) (from 0.1 to 0.026) on TruthfulQA Llama3, and a **44% reduction**, from 0.09 to 0.05, on TruthfulQA Vicuna. Similar improvements are observed across other datasets (as shown in Table 1 and Table 2), further demonstrating the effectiveness of our approach.
> >
> > Furthermore, our approach is highly efficient, utilizing only **1.7 million parameters**, in contrast to the state-of-the-art baseline Apricot, which relies on the much larger DeBERTa model with **86 million parameters**. This demonstrates our method’s efficiency and effectiveness.
> >
> > Thank you once again for your comments. We hope this clarifies and addresses your concerns.

---

### Official Review · Reviewer_qGFu · 2024-11-02

**Soundness:** 3
**Presentation:** 3
**Contribution:** 2
**Rating:** 3
**Confidence:** 5

**Summary:**

The paper presents a graph-based method for confidence calibration for LLMs, aiming to improve reliability by estimating response correctness through consistency. A graph neural network (GNN) is trained on a similarity graph of multiple LLM responses to a question, predicting correctness based on response consistency without processing language directly. Experimental results show substantial improvements over baseline methods in calibration performance and out-of-domain generalization.

**Strengths:**

1.	The topic of the paper is important. A well calibrated confidence of LLM outputs can benefit a lot of domains.
2.	The model exhibits strong generalizability across different datasets and LLMs, showing robustness against domain shifts, which is valuable for real-world applications.
3.	The paper includes sensitivity analyses and comparisons across various configurations, demonstrating the stability and effectiveness of the method under different setups and highlighting the performance benefits over established baselines.

**Weaknesses:**

1.	The requirement for sampling multiple responses and constructing similarity graphs for each query can introduce substantial computational costs, limiting scalability for real-time or resource-constrained applications.
2.	The motivation is not clear. First, how to choose the value of \tau in Eqn. is an extra challenge. Second, after having the correctness label, why do we need to leverage a GNN for classification? I think it is a typical text classification task. The motivation of choosing GNN is not clear.
3.	The reliance on Sentence-BERT embeddings for similarity computations could introduce biases and inconsistencies if embeddings do not fully capture semantic equivalence, potentially affecting calibration accuracy.

**Questions:**

refer to the weaknesses

---

> ### Author Response · Authors · 2024-11-24
>
> 1. **Regarding the computation**: We acknowledge that sampling multiple responses can introduce computational overhead. However, our sensitivity analysis reveals that the method performs well even with as few as 10 sampled answers (as shown in Figure 3b in the paper), significantly reducing computational costs without compromising performance. We have added further discussion about this.
> While computational cost is a limitation, our method offers a key advantage: it works for black-box models, requiring no access to the internal states or logits of the LLM. In contrast, other approaches—such as internal-state-based methods or logit-based methods—rely on white-box access to the model, making them less applicable to black-box scenarios. This broader applicability makes our method a viable alternative despite the computational trade-offs.
> 2. **Regarding the motivation for choosing GNN**: Our motivation is to leverage semantic similarity among answers to calibrate the answers’ confidence.
> Treating this task as a typical **text classification task**, where the question and answer are inputs to a model, would require a model with semantic understanding significantly stronger than the base LLM. Such an approach would require training a highly capable model, potentially exceeding the complexity and performance of the LLM itself, which is **impractical** in most scenarios.
>     **In contrast**, our graph-based approach offers a **simpler and more efficient solution** by focusing solely on the semantic similarity between answers. The GNN utilizes this similarity to aggregate information across clusters of answers, enabling effective classification without requiring a semantic understanding of individual answers. This design leverages the answer’s consistency pattern, avoiding the need for a more complex text classification model and making the task computationally feasible.
> **Regarding the choice of \tau** in Equation 4, \tau=0.3 is chosen following the previous work [1], which satisfies most cases.
>
> 3. We agree that there may be some bias in sentence-BERT embeddings for similarity computations. To test the robustness of various embedding strategies, we evaluated the sensitivity of our method to the quality of embeddings by experimenting with alternative embedding methods, such as Roberta and GPT3 embeddings. The results show that our approach remains robust across different embedding techniques, indicating that our method is not overly dependent on the specific nuances of Sentence-BERT.
>
> [1] Kuhn, Lorenz, Yarin Gal, and Sebastian Farquhar. "Semantic uncertainty: Linguistic invariances for uncertainty estimation in natural language generation." arXiv preprint arXiv:2302.09664 (2023).

---

> > ### Author Response · Authors · 2024-11-27
> >
> > Dear Reviewer qGFu, thank you once again for your valuable feedback.
> > We have added further discussions and conducted additional experiments, and we sincerely hope these address your concerns. At your convenience, we kindly ask if you could review our response. Your feedback is valuable to us, and we look forward to hearing any further thoughts or questions you may have.
> >
> > Kind regards, Authors

---

### Official Review · Reviewer_e4j6 · 2024-11-04

**Soundness:** 3
**Presentation:** 3
**Contribution:** 3
**Rating:** 5
**Confidence:** 5

**Summary:**

This paper considers the problem of producing meaningful uncertainty estimates given samples from a large language model. The approach relies on repeated sampling the LLM and forming semantically equivalent clusters of responses. These clusters are converted to a fully-connected similarity graph based on SBERT similarity between the responses contained in those clusters, which is the input to a supervised post-hoc calibration graph neural network (GNN). The node classification problem is to predict for each response if it is correct. The GNN does not take as input the SBERT embedding vectors to avoid dependency on textual information, but uses the cluster IDs and the similarity between each response. The clustering is obtained using the k-means algorithm. The appendix says that 30 responses are sampled and that K is set to 3 for k-means. The robustness of the approach is further (marginally) improved by forming multiple prompts for each question to increase the variety of response; further variety is associated with greater uncertainty. The proposed approach is compared to a diverse set of baseline methods on two standard QA tasks and is found to perform favourably in terms of calibration performance. The approach performs well with only small amounts of training samples.

**Strengths:**

* The paper considers an important problem (LLM confidence estimation).
* A diverse set of baseline methods are included (although I would have liked more information about how they were tuned compared to the baseline).
* The approach is applicable to “black box” models since it only relies on being able to sample from the model, rather than requiring access to the activations or full predictive distribution.
* The idea of using a GNN to calibrate a set of responses given a measure of semantic similarity between them is appealing.

**Weaknesses:**

* The use of a fixed number of samples and K for K-means seems like it could introduce some issues. If there’s no/little ambiguity won’t this still return three different clusters? At least some more sensitivity analysis should be done for K but this seems like a fundamental issue with the approach since choosing the number of clusters is underspecified in general.
* The quality of the approach seems like it depends strongly on the semantic similarity used. SBERT is quite an old approach and it’s unclear how well it generalises to various settings. I would have liked to see more analysis of the choice of semantic similarity.
* The evaluation is conducted on two public QA benchmark datasets from 2017 and 2019, which are almost certainly in the training dataset for many LLMs. How does this impact performance?
* There is no meaningful discussion of the limitations.
* Regarding the writing, it should be made more apparent what data is treated as "calibration data" vs. "validation data" vs. "test data"
* Overall, my biggest concern with the paper that the evaluation is limited to two fairly "easy" QA datasets. This makes it difficult to gauge how general the proposed approach is.

**Questions:**

See Weaknesses.

---

> ### Author Response · Authors · 2024-11-24
>
> We thank the reviewer's constructive comments and valuable suggestions of our work.
> 1. **Clarification of the k**: We thank the reviewer for pointing this out. To clarify, the maximum number of clusters is set to K = 3, but the algorithm naturally returns fewer clusters in cases of no or little ambiguity.  Note that the cluster memberships only provide one type of input feature to the GNN. The GNN also uses response similarities as edge features, so the GNN may give less weight to cluster memberships if they are noisy. We hope this alleviates the concerns about the specification of K  in our context.
> 2. **Regarding more datasets**: We have included experiments on TruthfulQA in Table 1 and Table 2 in the experiment Section. The experiment results are shown below. They provide further evidence of the effectiveness of our model:
> ### Table 1: Experimental results for TruthfulQA with Llama3 model
> |Method|AUROC|ECE|Brier|
> |----------|----------|----------|----------|
> |GraphSpectral [Lin et al.]|0.63|0.312|0.33|
> |GraphSpectral + iso calibration|0.63|0.091|0.19|
> |GraphSpectral + platt calibration|0.63|0.151|0.22|
> |Self-check GPT [Manakul et al.]|0.56|0.353|0.36|
> |Apricot[Ulmer et al.]|0.69|0.173|0.20|
> |Apricot + iso calibration |0.68|0.112|0.20|
> |Apricot + platt calibration|0.69|0.131|0.23|
> |**Ours**|**0.70**|**0.026**|**0.18**|
>
> ### Table 2: Experimental results for TruthfulQA with Vicuna model
> |Method|AUROC|ECE|Brier|
> |----------|----------|----------|----------|
> |GraphSpectral [Lin et al.]|0.64|0.226|0.286|
> |GraphSpectral + iso calibration|0.64|0.092|0.297|
> |GraphSpectral + platt calibration|0.64|0.183|0.307|
> |Self-check GPT [Manakul et al.]|0.55|0.308|0.28|
> |Apricot[Ulmer et al.]|0.76|0.11|**0.19**|
> |Apricot + iso calibration |0.76|0.09|**0.19**|
> |Apricot + platt calibration|0.76|0.08|0.20|
> |**Ours**|**0.77**|**0.05**|0.20|
> We hope this helps to alleviate your concern.
>
> 3. **Regarding the limitations**: We thank the reviewer for pointing this out. We’ve revised the limitations section in the conclusion accordingly.
> 4. **Clarification about the writing**:
> Thank you for pointing this out. We have clarified the distinctions between “calibration data,” “validation data,” and “test data” in the corresponding sections of the paper. The calibration data is the training data for our model, and validation data is a separate split used to tune the hyperparameters of the GNN calibrator. The test data is to test the performance of calibration capability.
>
> We appreciate the reviewer's careful reading and suggestions for this paper. We believe these revisions help improve the manuscript a lot.

---

> > ### Author Response · Authors · 2024-11-27
> >
> > Dear Reviewer e4j6, thank you once again for your valuable feedback.
> > We have added further discussions and conducted additional experiments, and we sincerely hope this will address your concerns. We kindly ask if you could take a moment to review our responses at your convenience. Your suggestions are super important to us, and we look forward to hearing any additional thoughts or questions you may have.
> >
> > Kind regards,
> > Authors

---

> ### Author Response · Authors · 2024-12-03
> **Thank you for your suggesstions**
>
> Dear reviewer e4j6,
>
> We thank you once again for your valuable suggestions. We’d like to point out that we have conducted additional experiments on the more challenging TruthfulQA dataset (the accuracy is 56% on llama3). And our method demonstrates much better performance than the baselines. Specifically, our method achieves a 74% reduction in Expected calibration error (ECE) (from 0.1 to 0.026) on TruthfulQA with Llama3, and a 44% reduction, from 0.09 to 0.05, on TruthfulQA with Vicuna. This is generally due to the non-dependency on textual information. The results show more evidence of our method’s effectiveness.
>
> If there are any remaining questions, we would be happy to provide additional clarification.
>
> Kind regards, Authors

---

### Official Review · Reviewer_fhWK · 2024-11-05

**Soundness:** 2
**Presentation:** 2
**Contribution:** 2
**Rating:** 6
**Confidence:** 3

**Summary:**

To enhance the calibration performance of LLMs, this paper proposes to combine the LLM’s self-consistency with labeled data and train an auxiliary model to estimate the correctness of its responses to questions. Experiments demonstrate that the proposed method outperforms baselines in confidence calibration on two datasets.

**Strengths:**

1.	The paper presents a novel method to confidence calibration in large language models by leveraging graph neural networks and the consistency among multiple model responses.

2. The manuscript is well-organized and the experiments are comprehensive.

**Weaknesses:**

1. The premise proposed in this paper is if LLMs give similar response. Then there is less uncertainty, and these responses tend to have a high probability of being correct.  It is common to use the idea of self-consistency to guide the selection of the most appropriate answer from multiple answers, but it may not be reasonable to use it as a probability to evaluate the correctness of an answer. For example, the LLM may exhibits generate the same wrong answer when sampling multiple answers. In this case, is the proposed method still effective?

2. This paper trains another model to calibrate the correctness of the answers generated by LLM. What is the difference between this auxiliary model and the "reward model" in the LLM evaluation work, which is used to evaluate the quality of responses? If there is a difference, the author should point out the difference with this type of work; if it is similar, the author should add a description of these works and performance comparison in related work and experiments.

3. This work uses semantic similarity to cluster different answers. Even so, answers within the same cluster may still be completely different answers. Because they just exhibit the similar reasoning paths but gives the different final answers. This phenomenon is also mentioned in paper [1]. But according to Equation 3, the author calculates the correctness probability of each cluster using the ratio of numbers. Why? How to explain the reasonability?
[1] SaySelf: Teaching LLMs to Express Confidence with Self-Reflective Rationales

4. In Method Section, the details how the training dataset was created are unclear. For example, for the same question, will the confidence information of each cluster be input into the GNN? It is recommended that the author add a paragraph to describe the complete process of constructing GNN training data.

5. This paper constructs training data based on the results of self-consistency. The author should add a comparison with the performance of the simplest self-consistency method to estimate confidence in the baselines.

6. Section 4.3 lacks analysis of DDL results.

**Questions:**

1.	Are the confidence values of answers in the same cluster the same?

2.	When verifying the ability of the proposed method on OOD in Section 4.3, I recommend that the author add other datasets of tasks that are quite different from the training data to verify the generalization ability, such as Math

---

> ### Author Response · Authors · 2024-11-24
>
> We thank the reviewer's constructive comments and valuable suggestions of our work.
>
> 1. **Regarding the assumption**: Our assumption is that higher consistency among LLM’s responses indicates a higher probability of correctness. To validate this, we conducted an empirical evaluation on the TriviaQA dataset. On the TriviaQA dataset, we observed a **strong statistical correlation between correctness and consistency with R² = 0.966**. This suggests that statistically speaking, responses exhibiting greater consistency are more likely to be correct. We further validated this assumption using the TruthfulQA dataset by analyzing the proportion of highly consistent yet incorrect answers. Results showed that only 13% of all incorrect responses exhibited high consistency.
> Similarly, this trend was observed in other models, such as GPT-4-o1, on the SimpleQA dataset, where more consistent answers demonstrated higher accuracy (see Figure 2 in [1]).  While this confirms the general reliability of our assumption, we acknowledge that there are always outliers where the LLM generates consistent but incorrect responses due to systematic biases or errors. While these cases exist, they are statistically rare, and the strong correlation observed in our study supports the utility of consistency as a proxy for correctness in most scenarios. We will include a detailed discussion in the appendix. Furthermore, this assumption aligns with insights from prior work [1][2], further reinforcing its validity as a practical method for calibration.
>
> 2. **Regarding the difference of reward model**:The primary difference lies in the **goal of the auxiliary model versus the reward model**. The reward model is designed to provide feedback to LLMs to help them align with human preferences, focusing on attributes like honesty, helpfulness, and harmlessness. Specifically, reward models are trained to mimic human preferability by ranking responses based on human feedback, and they are used to fine-tune language models via reinforcement learning (e.g., RLHF). These models aim to optimize for preference, not confidence.
> In contrast, our auxiliary model is focused on confidence calibration—determining the probability that a generated answer is correct. This distinction is crucial, as preference and correctness are not synonymous. For example, a response may align with human preferences but still be factually incorrect. Our work addresses this gap by calibrating confidence, not preference.
> Further differences include:
>     **Neural Network Architecture**: Reward models typically use language models like BERT, RoBERTa, or GPT to process the semantic content of responses and predict human preferences. These architectures are parameter-heavy and computationally intensive.
> Conversely, we employ a Graph Neural Network that focuses on the semantic similarity among responses. This allows us to efficiently calibrate confidence scores among multiple responses without requiring the deeper contextual understanding needed for preference modeling.
>     **Loss Function**: Reward models use a ranking loss, which encourages higher-ranking probabilities for responses with greater preference alignment. In contrast, our model is trained with a loss function specifically designed for confidence calibration, ensuring the model predicts accurate confidence scores based on the consistency among answers.
> We will incorporate a discussion of the works on reward models, such as [3] and [4], in the related work section, clarifying these distinctions and providing a comprehensive context for the contributions of our approach.
>
>
> [1] Wei, Jason, et al. "Measuring short-form factuality in large language models." arXiv preprint arXiv:2411.04368 (2024).
> [2] Lin, Zhen, Shubhendu Trivedi, and Jimeng Sun. "Generating with Confidence: Uncertainty Quantification for Black-box Large Language Models." Transactions on Machine Learning Research.
> [3] Ouyang, Long, et al. "Training language models to follow instructions with human feedback." Advances in Neural Information Processing Systems, 35 (2022): 27730-27744.
> [4] Stiennon, Nisan, et al. "Learning to summarize with human feedback." Advances in Neural Information Processing Systems, 33 (2020): 3008-3021.

---

> ### Author Response · Authors · 2024-11-24
>
> 3. Through our experiments, we observed that samples within the same cluster tend to share the same meaning, even when expressed using different wording or reasoning paths. In the TriviaQA dataset, 91.4% of answers within a cluster converged to the same final answer. This strong alignment indicates that clustering based on semantic similarity serves as a reasonable proxy for grouping similar answers.
> Additionally, [5] reported a similar observation. Their study highlighted that responses within a cluster often share reasoning paths or underlying semantics, despite minor differences in expression. This further supports the rationale for aggregating answers within clusters. While we acknowledge the existence of outliers, the high consistency observed within clusters across datasets and corroborated by related work demonstrates the practicality and effectiveness of this approach.
>
> 4. We appreciate the reviewer's valuable feedback. For each question, the process of constructing the GNN training data is as follows:
> **Sampling Answers**: We generate multiple answers for each question using multinomial sampling to capture a diverse range of possible responses from the LLM.
> **Constructing the Semantic Similarity Graph**: A semantic similarity graph is created where each node represents an answer, and edges between nodes represent the semantic similarity between answers. The similarity is computed using a predefined metric.
> **Node Features**: Each node is assigned features that include the cluster ID, which groups semantically similar answers. This enables the GNN to aggregate information within and across clusters during training.
> We have provided an explanation in lines 213-232 of the Method Section. We will refine this section to include a more detailed description.
> 5. We thank the reviewer’s feedback. We would like to point out that the baseline method, Graph Spectral, utilizes a self-consistency mechanism to estimate uncertainty, but its confidence scores are not calibrated. To ensure a fair comparison, we enhance the baseline by applying Platt Scaling to calibrate its confidence estimates with the same training dataset to our approach, resulting in the Graph-Spectral + Platt Scaling baseline. This allows us to directly compare the effectiveness of our approach against a calibrated version of the self-consistency method. We have already included this comparison in the results section for a comprehensive evaluation.
>
> [5] Xu, Tianyang, et al. "SaySelf: Teaching LLMs to Express Confidence with Self-Reflective Rationales." arXiv preprint arXiv:2405.20974 (2024).

---

> > ### Author Response · Authors · 2024-11-27
> >
> > Dear Reviewer fhWK, thank you once again for your valuable feedback.
> > We have added further discussions, and we sincerely hope this will address your concerns. We kindly ask if you could take a moment to review our responses at your convenience. Your suggestions are super important to us, and we look forward to hearing any additional thoughts or questions you may have.
> >
> > Kind regards,
> > Authors

---

> > > ### Comment · Reviewer_fhWK · 2024-12-02
> > >
> > > Part of my concerns have been clarified. The score will be increased.

---

> > > > ### Author Response · Authors · 2024-12-03
> > > >
> > > > Thank you for your comments and for considering an increase in the score.
> > > >
> > > > We are glad to hear that part of your concerns have been clarified. If there are any remaining questions or aspects of the work that could benefit from further elaboration, we would be happy to provide additional clarification. We appreciate your thoughtful review.
> > > >
> > > > We also wanted to add more discussions for your questions to provide additional clarity and address your concerns more comprehensively.
> > > >
> > > > Q1: The confidence values of answers within the same cluster do not need to be identical because the edge features vary. For two answers in the same cluster, their edge weights to other answers (nodes) differ, as the edge weights are computed based on the similarity of node embeddings. This variation affects the GCN message-passing process, where edge weights are utilized to adjust the aggregation of neighboring information. This also ensures that the cluster feature plays a less important role if they are noisy.
> > > >
> > > > Q2: We appreciate the reviewer’s valuable suggestions. We have included TruthfulQA, which represents a distinct OOD scenario. This dataset is more challenging compared to TriviaQA, as TriviaQA focuses on quiz-style QA that emphasizes broad knowledge retrieval, while TruthfulQA is specifically designed to test a model’s ability to reason about the truthfulness of answers.
> > > >
> > > > Thank you once again for your comments. We hope this helps clarify and address your concerns.

---

### Author Response · Authors · 2024-12-04

Dear Reviewers and ACs,
We would like to thank all the insightful comments and valuable suggestions provided by all reviewers. In the following, we would like to summarize the contributions and revisions of this paper again.

**Contributions**:

**[High-level insight]** This paper proposes a graph-based confidence calibration method for LLMs that uses a similarity graph to encode relationships among multiple responses to the same question. By employing a GNN to calibrate response correctness, the method significantly improves confidence calibration performance and generalization to out-of-domain data compared to baseline approaches. This method is novel and appealing (*R fhWK*, *R e4j6*, and *R RPHJ*), has strong performance and generalizability (*R fhWK*, *R e4j6*, *R RPHJ*, and *R qGFu*), and is applicable to “black box” models (*R e4j6*).

**Responses and Revisions**

- **[For reviewer fhWK]**
  - **Clarifications:** We have provided a detailed analysis in the response, and it resulted in a positive adjustment in scoring.
  - **Extended experimental analysis:** Additional experiments have been conducted on more challenging TruthfulQA datasets and demonstrate the generalizability of our method to diverse datasets.
- **[For reviewer e4j6]**
  - **Clarification on hyperparameters:** We have added more details on the choices of hyperparameters.
  - **Additional experiments:** Additional experiments have been conducted on the more challenging TruthfulQA dataset and demonstrated better performance.
  - We hope our further experiments and clarification have been helpful.
- **[For reviewer qGFu]**
  - **Regarding the Computation:** We conducted a sensitivity analysis on the impact of the number of sampled answers, and the results demonstrate that the approach maintains strong performance even with fewer sampled answers. Additionally, computational efficiency can be improved through parallel sampling.
  - **Motivation for choosing GNN:** We have provided a detailed analysis of why we chose the GNN in the responses. Our method utilizes the topology of the similarity graph and does not rely on the semantic meaning. Furthermore, by utilizing the GNN’s message-passing mechanism, it can gradually integrate information across the entire graph, resulting in improved calibration.
We hope our further clarification has been helpful.
- **[For reviewer RPHJ]**
  - **Additional experiments:** We conducted additional experiments on the more challenging TruthfulQA dataset, which demonstrated improved performance and led to an increase in scoring.
  - **Figure Clarity:** We have revised Figure 1 to provide the intuition as to why the proposed method chose to use GNNs.

Thanks again for your efforts in reviewing our work, and we hope our responses can address any concerns about this work.
Thanks, Authors

---

### Note · Authors · 2025-01-22

I have read and agree with the venue's withdrawal policy on behalf of myself and my co-authors.